# Total Neoadjuvant Therapy in Localized Pancreatic Cancer: Is More Better?

**DOI:** 10.3390/cancers16132423

**Published:** 2024-06-30

**Authors:** Rita Saúde-Conde, Benjelloun El Ghali, Julie Navez, Christelle Bouchart, Jean-Luc Van Laethem

**Affiliations:** 1Digestive Oncology Department, Hôpitaux Universitaires de Bruxelles (HUB), Université Libre de Bruxelles, 1070 Brussels, Belgium; jean-luc.vanlaethem@hubruxelles.be; 2Department of Radiation Oncology, Hôpitaux Universitaires de Bruxelles (HUB), Institut Jules Bordet, Université Libre de Bruxelles (ULB), 1070 Brussels, Belgium; benjelloun.el.ghali@hubruxelles.be (B.E.G.); christelle.bouchart@hubruxelles.be (C.B.); 3Department of Abdominal Surgery and Transplantation, Hôpitaux Universitaires de Bruxelles (HUB), Hopital Erasme, Université Libre de Bruxelles, 1070 Brussels, Belgium; julie.navez@hubruxelles.be

**Keywords:** total neoadjuvant therapy, neoadjuvant therapy, radiotherapy, pancreatic cancer, borderline resectable, locally advanced

## Abstract

**Simple Summary:**

Pancreatic cancer is challenging to treat due to late diagnosis and limited options. Surgery, the main treatment, often leads to poor long-term outcomes, prompting exploration of alternative approaches like neoadjuvant therapy (NAT) and total neoadjuvant therapy (TNT). While NAT aims to improve surgical success and overall survival, there is room for enhancement, leading to consideration of neoadjuvant strategies combining full-dose chemotherapy and radiotherapy in TNT. TNT, integrating chemotherapy and radiotherapy before surgery, could increase the likelihood of successful surgery and cure for locally advanced cases. This article explores different TNT strategies, categorized based on radiotherapy techniques, offering a thorough analysis of their effectiveness in borderline resectable and locally advanced pancreatic cancer. The central question remains: does more treatment before surgery always yield better results?

**Abstract:**

Pancreatic ductal adenocarcinoma (PDAC) poses a significant challenge in oncology due to its advanced stage upon diagnosis and limited treatment options. Surgical resection, the primary curative approach, often results in poor long-term survival rates, leading to the exploration of alternative strategies like neoadjuvant therapy (NAT) and total neoadjuvant therapy (TNT). While NAT aims to enhance resectability and overall survival, there appears to be potential for improvement, prompting consideration of alternative neoadjuvant strategies integrating full-dose chemotherapy (CT) and radiotherapy (RT) in TNT approaches. TNT integrates chemotherapy and radiotherapy prior to surgery, potentially improving margin-negative resection rates and enabling curative resection for locally advanced cases. The lingering question: is more always better? This article categorizes TNT strategies into six main groups based on radiotherapy (RT) techniques: (1) conventional chemoradiotherapy (CRT), (2) the Dutch PREOPANC approach, (3) hypofractionated ablative intensity-modulated radiotherapy (HFA-IMRT), and stereotactic body radiotherapy (SBRT) techniques, which further divide into (4) non-ablative SBRT, (5) nearly ablative SBRT, and (6) adaptive ablative SBRT. A comprehensive analysis of the literature on TNT is provided for both borderline resectable pancreatic cancer (BRPC) and locally advanced pancreatic cancer (LAPC), with detailed sections for each.

## 1. Introduction

Pancreatic ductal adenocarcinoma (PDAC) poses a formidable challenge in oncology, marked by a grim prognosis primarily attributed to frequent distant metastases and/or advanced local growth precluding surgical intervention [1]. Pancreatic cancer (PC) ranks fourth in number of cancer deaths, with projections placing it at the second position by 2030 in the Western world [2]. Early detection is hindered by subtle and late-presenting symptoms, with close to half of patients already metastatic at diagnosis. Merely 20% of cases are immediately resectable, while approximately half of them fall into the category of “borderline resectable” (BR) and 1/3 are locally advanced (LA) and unresectable, according to the tumor’s relationship with adjacent vascular structures and the probability of obtaining a negative microscopic margin (R0) [2,3,4].

For all localized stages, surgical resection stands as the only potentially curable treatment, necessary to obtain long-term survival. Yet even in operated cases, the prognosis remains grim, with a 5-year overall survival of only 20%. While acknowledging that R0 status is one of the most important prognostic factors, it is worth noting that some patients with positive margins or margins < 1 mm survive without recurrence, highlighting the complexity of pancreatic cancer management [2,4]. As a standard practice, surgical intervention is commonly followed by adjuvant therapy, a strategy consistently proven to significantly improve survival rates. However, one should recognize that only half of the patients successfully complete the entire adjuvant therapy sequence [5,6,7]. As such, the conventional “surgery-first” approach has become less appealing, especially for individuals with BRPC and LAPC, where achieving a margin-negative resection is highly challenging [8]. Given the likelihood of micrometastatic disease at diagnosis, even in cases of localized disease, there has been emerging emphasis on the concept of neoadjuvant therapy (NAT) as a new therapeutic strategy to improve the probability of R0 resection and OS, in both resectable and BRPC. Similarly, a “sandwich” approach has gained attraction. This approach ensures the neoadjuvant administration of systemic therapy, minimizing the risk of missing a critical window for surgery and guaranteeing the delivery of systemic therapy, even in the presence of perioperative complications [9]. More recently, similarly to treatment strategies applied in rectal cancer, another concept called total neoadjuvant therapy (TNT) was developed by adding neoadjuvant radiotherapy (RT) to the treatment of BRPC and LAPC [10,11,12]. In TNT, the sequencing of chemotherapy (CT) and radiotherapy treatments has demonstrated an increased likelihood of patients achieving margin-negative and node-negative disease, coupled with higher rates of complete and major pathologic response which could lead to improved overall survival (OS) [5,13,14]. This novel approach in pancreatic cancer can mirror successful paradigms seen in rectal cancer, emphasizing the potential benefits of tailoring treatment sequences to enhance therapeutic outcomes and address challenges associated with completing postoperative adjuvant systemic therapy [15,16,17,18,19].

In parallel, efforts have been made by the American Joint Committee on Cancer (AJCC) and the American Hepato-Pancreato-Biliary Association (AHPBA), later adopted by the National Comprehensive Cancer Network (NCCN) in order to obtain a standardized definition of resectability [20,21,22,23]. While there is now a standardized definition of resectability criteria available, the current imperative lies in fostering a worldwide consensus allowing data comparison across clinical trials. In fact, many trials encompass a spectrum of pancreatic cancer patients, including and mixing those classified as resectable pancreatic cancer (RPC), BRPC, and even LAPC. Furthermore, it is paramount to prioritize the acquisition of a thorough understanding of the precise sequence, duration, and regimens of TNT that offer benefits to patients with BRPC and LAPC. In this review, we examine the existing evidence concerning the use of NAT for PDAC patients as well as an exploration of the advantages, drawbacks, and limitations of NAT. Thereafter, we delve into the available evidence related specifically to the different TNT strategies available for localized PDAC and compare their aspects regarding chemotherapy regimen/exposure/duration, RT techniques, and treatment sequences. Lastly, we will undertake a review of the assessment of treatment response and selection of patients in the era of precision medicine.

## 2. What Is Total Neoadjuvant Therapy?

TNT strategy is gaining popularity as it offers the potential advantage of increasing the likelihood of achieving negative margins as well as the potential to convert LAPC into candidates for curative oncological resection. TNT specifically aims to utilize a combination of CT and RT with or without concomitant chemotherapy before surgical exploration [5,24,25,26]. 

In this article, we divided the TNT strategies into six main categories according to the RT technique used: (1) conventional chemoradiotherapy (CRT) approach; (2) Dutch PREOPANC approach, (3) hypofractionated ablative intensity modulated radiotherapy (HFA-IMRT) strategy, and finally stereotaxic body radiotherapy (SBRT) techniques. The latter can be divided into the following: (4) non-ablative SBRT (biological effective doses (BED) 10 around 50–60 Gy), (5) nearly ablative SBRT (BED10 around 70–85 Gy), and finally (6) adaptive ablative SBRT approach (BED10 around 100 Gy). A comprehensive analysis of the existing literature on TNT was conducted for both BRPC and LAPC patients, detailed in distinct sections.

## 3. General Risks and Benefits of NAT and TNT

While NAT has gained increasing popularity, it is also associated with potential disadvantages. The risk of distant metastases development during NAT can be as high as 30% [27,28]. This rate aligns with the early disease recurrence observed in patients with RPC who undergo immediate surgery and can be regarded as analogous to an early recurrence (within 6 months) following surgical resection after NAT [29,30]. However, the risk of disease progression during NAT should be analyzed as part of selecting optimal surgery candidates [30]. Patients with PDAC encountering early distant metastases after undergoing upfront major pancreatectomy likely undergo surgery-related burdens without oncologic benefits [31]. On the other hand, delayed surgery, whether due to treatment toxicity or patient’s deterioration, could affect outcomes. Yet, studies suggest this delay in surgery may also allow for careful patient selection and preoperative preparations, such as pre-rehabilitation and nutritional optimization. Two retrospective cohort studies from the US National Cancer Database explored the impact of waiting time before surgery [32,33]. Mirkin et al. found that long-term survival remained unaffected by the waiting time [33]. Swords et al. observed lower mortality rates for patients with average and long waiting times for surgery. Furthermore, the 90-day mortality was lower in both groups compared to shorter waiting time for surgery [32].

On the other hand, postoperative complications may hinder the administration of adjuvant therapy associated with improved outcomes [34,35]. A retrospective study found that in the NAT setting, 83% of patients completed the entire treatment sequence, versus 58% in the surgery-first group. For the surgery-first group, the reasons for the failure to complete the entire sequence included early disease progression, major complications, and deterioration of performance status. Completing the entire sequence led to significantly longer overall survival (36.0 versus 11.0 months, *p* < 0.001) [36]. Additionally, NAT does not seem to increase the postoperative complication rate [37,38]. A systematic review found comparable rates of fistula and infection, as well as similar mortality [37]. Additional studies noted reduced specific complications such as pancreatic fistula and hemorrhage or gastroparesis after NAT [39,40]. The most plausible explanation for these observations is that NAT, particularly RT, may induce pancreatic fibrosis, leading to a decreased fistula incidence and lower complication risks [30,38]. Kim et al. studied specifically TNT in operable PDAC patients, analyzing 541 patients (58% BPRC) [8]. Of these, 16% received TNT, and 84% received standard neoadjuvant treatment (SNT). Treatment duration was longer in the TNT group (5.5 months) compared to SNT (4 months) (*p* < 0.01), with similar completion rates for intended therapy and subsequent surgery (*p* = 0.90). The study suggests that patients tolerating SNT might benefit from TNT without compromising surgical operability. Accordingly, the results from the randomized phase III PREOPANC-2 trial comparing NAT FOLFIRINOX versus TNT for RPC/BRPC showed no differences in resection rates (*p* = 0.69) [41].

Patients undergoing neoadjuvant chemotherapy may benefit from changing their chemotherapy regimen or adding radiotherapy before resection in case of early loco-regional progression or no relevant response [7,42]. TNT may facilitate such a therapeutic switch. Alva-Ruez et al. reported the Mayo Clinic’s experience with chemotherapy switching in 468 patients with BRPC and LAPC [42]. Following a median of six cycles of neoadjuvant chemotherapy, 30% underwent a switch due to progression, absence of an objective response, or toxicity. Of those, 72% were resected. Although acknowledging probable patient selection bias, this strategy presents an intriguing prospect for future trials comparing SNT vs. TNT [31]. 

Overall, the potential benefits of NAT and TNT appear to outweigh their drawbacks.

## 4. The Advent of TNT for BRPC Patients

While uncertainty remains, whereas neoadjuvant treatment should be given to RPC patients, especially following the results of the NORPACT-1 trial, the results from ongoing randomized trials are awaited [43,44]. On the other side, NCCN and ESMO guidelines recommend the use of NAT in the management of BRPC patients where the neoadjuvant approach appears to be more beneficial [45,46]. 

### 4.1. Use of Single Modality NAT

In a recent meta-analysis, the Dutch Pancreatic Cancer group compared upfront surgery with the use of a single NAT modality for both RPC and BRPC [47]. They analyzed 38 studies, involving 3884 patients, finding that the median overall survival (OS) was longer in the NAT group (19.2 months versus 12.8 months in the upfront-surgery group). The NAT group included the use of various chemotherapy or RT techniques, leading to heterogeneity that precluded subgroup analysis [47]. A separate patient-level meta-analysis specifically focused on neoadjuvant FOLFIRINOX in 313 BRPC patients, showing a resection rate of 67.8%, and an R0 resection rate of 83.9% (95% CI: 76.8–89.1) [48]. The median OS was 22.2 months, with a median progression-free survival (PFS) of 18.0 months [48].

To date, only two prospective randomized trials specifically explored the value of single modality NAT in patients with BRPC only [49,50]. A phase II Korean trial comparing preoperative versus postoperative CRT showed a higher resection rate (51.8% versus 26.1%) and 2-year survival rate (40.7% versus 26.1%; HR, 1.495; 95% CI, 0.66–3.36; *p* = 0.028), in favor of the NAT group [49]. The recurrence rate was notably high in both groups (around 88%), but the median OS was higher in the NAT group (21 versus 12 months). However, this trial was closed prematurely for efficacy following the first interim analysis, with only 58 enrolled patients [49]. In the randomized phase II ESPAC5 trial, 90 BRPC patients were assigned to receive either immediate surgery, 8 weeks of neoadjuvant gemcitabine/capecitabine, neoadjuvant FOLFIRINOX, or capecitabine-based CRT (total dose 50·4 Gy in 28 daily fractions) [50]. While resection rates did not significantly differ between immediate surgery and NAT groups (68% vs. 55%), R0 resection rates were indeed higher in the NAT groups (23% vs. 14%, *p* = 0.49). One-year OS rates showed significant differences, with higher rates in favor of NAT groups (39% for immediate surgery, 78% for gemcitabine plus capecitabine, 84% for FOLFIRINOX, and 60% for CRT, *p* = 0.0028) and DFS compared to immediate surgery (*p* = 0.016) [50].

Three additional randomized clinical trials (RCTs), including both BRPC and RPC patients, have also explored single modality NAT with distinct treatment regimens [51,52,53]. Caution is necessary when analyzing these trials due to the restricted number of patients in two of them, the limited accessibility of data in a third, and the variability in regimen treatments used across the trials. In 2015, Golcher et al. conducted a phase II RCT evaluating the role of NAT CRT (55.8 Gy in 31 fractions, gemcitabine/cisplatin-based) versus surgery first. However, the trial was prematurely closed due to poor patient enrolment (n = 73). The partial results showed no significant differences in R0 resection (48% vs. 52%, *p* = 0.81), postoperative complications, or survival in the intention to treat analysis [51]. In 2021, the phase II/III RCT NEPAFOX comparing adjuvant gemcitabine and perioperative FOLFIRINOX was also halted due to recruitment challenges after enrolling 40 RPC and BRPC patients [53]. Among the patients, 57.9% did not receive surgery due to progressive disease, death, toxicities, and pneumonia. Perioperative morbidity was higher in the surgery-first group (72.2%), but R0 resection rate was higher (72.2%) compared to 45.5% for both in the NAT group. Median OS was also favored in the surgery-first group (25.68 months vs. 10.03 months, HR 0.722, *p* = 0.4099) [53]. The Japanese phase III RCT Prep-02/JSAP-05 evaluated the role of two cycles of NAT using gemcitabine and S-1, followed by 6 months of adjuvant chemotherapy in 362 RPC and BRPC patients. While no differences were seen between R0 resection rates and morbidity of surgery (no statistical data were provided in the abstract), the median OS was higher in the NAT group compared with the upfront-surgery group (36.7 vs. 26.6 months, HR 0.73, 95%CI 0.55–0.94, *p* = 0.015) [54]. 

While these findings support an NAT strategy, they also indicate the possibility of improvement of the NAT sequences in order to allow for better outcomes for BRPC patients. This prompts consideration of alternative neoadjuvant strategies combining full-dose CT and RT in TNT approaches.

### 4.2. Use of TNT Strategies for BRPC

#### 4.2.1. Conventional CRT with Induction or Consolidation Chemotherapy Prior to Surgery

Similarly, to a TNT approach often used in rectal cancer patients (OPRA trial), an approach that has been used in localized PDAC patients consists in a neoadjuvant conventional CRT (BED10 around 60 Gy) either preceded by induction chemotherapy or followed by consolidation chemotherapy [55]. 

Only two non-randomized phase II trials addressed the question of whether induction chemotherapy followed by CRT is beneficial specifically for BRPC patients [13,56]. First, Katz et al. presented the results of 22 patients with BRPC (intergroup criteria) who underwent an induction chemotherapy treatment of four cycles of FOLFIRINOX followed by CRT with concomitant capecitabine. The results showed a resection rate of 68% and an R0 resection rate of 93%, with 13% achieving a pathological complete response (pCR). The median OS was 21.7 months, indicating the feasibility of this TNT regimen despite notable toxicity associated with preoperative therapies (grade G ≥ 3: 64%) [13]. Second, Nagakawa et al. studied an induction chemotherapy treatment consisting of two cycles of gemcitabine followed by chemoradiotherapy with gemcitabine and S1 in 27 patients with BRPC (author’s own definition derived from NCCN). While in this study 48.1% of patients had ≥G3 adverse events (AEs) (mostly hematologic), a similar resection rate (70.3%) and R0 resection rate (94.7%) were found. Thirteen patients (68.4%) experienced distant metastasis at the initial site of recurrence following resection. Local recurrence occurred in just one of these patients (7.7%). The median OS was 22.4 months, with a corresponding 1-year survival rate of 81.3% [56]. The limited patient cohort and the lack of comparative analysis in both trials hinder the formulation of definitive conclusions in BPRC. Yet, retrospective studies encompassing a larger patient cohort, as outlined in Table 1, have shown encouraging results [6,57,58,59]. However, the use conventional CRT, associated with the delivery of low BED10 to the tumor and prolonged interruption of full-dose chemotherapy, may not be the fittest approach in PDAC, known to be radioresistant. Therefore, other TNT approaches were developed and studied. 

#### 4.2.2. The Dutch TNT Strategy: PREOPANC Regimen

The “PREOPANC approach” established by the Dutch Pancreatic Cancer Group consists in the administration of preoperative gemcitabine associated with hypofractionated CRT (36 Gy in 15 fractions; three cycles of gemcitabine 1000 mg/m^2^—second cycle concomitantly with CRT) [41,67,73]. The doses of gemcitabine and the RT scheme were established according to two phase I/II studies in order to use also full-dose gemcitabine during CRT [74,75].

The phase III RCT PREOPANC-1 (Table 1) compared the Dutch regimen versus upfront surgery in 246 RPC and BRPC (46%). Adjuvant gemcitabine was administered for six cycles in the upfront-surgery group and for four cycles in the CRT group. After long-term follow-up (59 months), the results highlighted a superior R0 resection rate (72% versus 43%, *p*  <  0.001), median loco-regional failure-free interval (13.4 months versus 31.2, *p*  =  0.004), and notably lower occurrences of pathologic lymph nodes (35% versus 82%, *p*  <  0.001) and perineural (45% versus 85%, *p*  <  0.001) and vascular invasion (36 versus 65%, *p* < 0.001) in favor of the TNT group. The median OS in the intention-to-treat population, which was the primary endpoint, was modestly improved for the CRT arm (15.7 months versus 14.3 months, *p* = 0.025), as well as distant metastasis-free survival (17.4 months versus 12.5 months, *p* = 0.070) and median DFS (8.1 versus 7.7 months, *p* = 0.09). Interestingly, the TNT group achieved a 5-year overall survival rate of 20.5%, whereas the upfront-surgery group attained a rate of 6.5%. For BRPC patients, the unstratified HR was 0.67 (95% CI 0.45–0.99), favoring TNT in OS, but did not reach statistical significance (*p* = 0.88). However, emerging multi-agent chemotherapy protocols like gemcitabine/nab-paclitaxel, FOLFIRINOX (FFX), and more recently NALIRIFOX have exhibited encouraging advancements compared to gemcitabine in metastatic, adjuvant, and increasingly in the neoadjuvant strategy as well [76,77,78]. Although FOLFIRINOX appears to be an efficacious option, uncertainty remains on its superiority compared to alternative multi-agent chemotherapy regimens [79,80,81,82]. As such, the phase III PREOPANC-2 trial (Table 1) compared the Dutch TNT regimen versus neoadjuvant FFX (eight cycles prior to surgery) [41]. Recently presented results from 375 randomized RPC and BRPC patients showed a median OS of 21.9 months in the FFX group versus 21.3 months in the TNT arm (HR 0.87; 95% CI 0.68–1.12, *p* = 0.28). Completion rates (*p* < 0.001) and resection rates were similar between both groups (77% in the FFX group and 75% in the TNT group, *p* = 0.69), as well as R0 resection rates (61% and 67% in the FFX and TNT group, respectively, *p* = 0.28). Serious adverse events (SAEs) were also similar (49% in the FFX group and 43% in the TNT arm, *p* = 0.26). While no differences were seen in OS, patients treated with the PREOPANC approach had a significantly higher pathologic response in lymph nodes (ypN0: 7% in the FFX group and 58% in the TNT arm; ypN2: 20% in the FFX group vs. 7% in the TNT arm, *p* < 0.01), hinting at the potential role of radiotherapy. Of note, BRPC patients were a minority in both groups (around 35%), suggesting the need for separate future studies to address each population. Furthermore, it is important to note that no adjuvant treatment was performed in the FFX arm contrary to the TNT arm (four cycles of adjuvant gemcitabine) and only 62% of the patients in the FFX arm completed the whole treatment, therefore leaving the question of potential further outcomes improvement in case of the use of a more optimized FOLFIRINOX arm.

Therefore, despite several interesting results with the Dutch TNT regimen, the use of gemcitabine monotherapy—the old chemotherapy standard for PDAC—also leaves room for further improvement of the TNT strategy.

#### 4.2.3. Stereotactic Body Radiotherapy (SBRT)

SBRT enables the precise administration of elevated doses to tumors over a limited number of sessions (1–5), thereby minimizing dose exposure and toxicity to nearby organs at risk (OARs) [83]. Unlike conventional RT, SBRT avoids irradiating large volumes and does not include prophylactic irradiation of neighboring lymph node areas. With its shorter treatment duration (1 week versus 4–6 weeks), patients undergoing SBRT can promptly resume systemic therapy, minimizing interruptions to full-dose chemotherapy [4,84]. Furthermore, SBRT aims to enhance local control by delivering higher BED to tumors, addressing the significant proportion of PDAC patients who succumb to local progression alone [85]. Subsequently, there has been a surge in interest regarding the application of SBRT in localized PDAC patients. The incorporation of SBRT into a neoadjuvant strategy presents a seamless opportunity to deliver particularly high BED on tumor-vessel interfaces (TVIs), areas where tumors come into contact with blood vessels. By targeting precisely these regions, there is heightened potential to optimize oncological (R0) resection rates [4,86]. However, prospective data remain scarce [85].

SBRT techniques can be divided into non-ablative, nearly ablative, and ablative regimen.

##### Non-Ablative SBRT

Protective non-ablative SBRT schemes are mainly comprised between 25 and 35 Gy in five fractions, resulting in a maximal BED10 of 60 Gy, which falls far below the ablative doses that could be delivered with an SBRT technique [4]. If treatment safety is assured, the oncological efficacy of non-ablative SBRT is unsurprisingly lacking. The results of the main phase II trial and retrospective study are resumed in Table 2 [87,88,89,90,91,92,93,94]. This was illustrated with the results of the randomized phase II Alliance A021501 trial, comparing outcomes in BRPC patients who underwent induction FFX alone versus those who received TNT including FFX followed by non-ablative SBRT (33 Gy in five fractions with SIB up to 40 Gy at TVI or 25 Gy in five fractions) [95]. Initially, this trial encompassed three arms; however, following the findings of the LAP07 trial, the arm involving FFX followed by conventional CRT was terminated. The trial was halted after an interim analysis of 30 patients revealing a crossing of the futility boundary for R0 resection rates in the TNT arm (less than 11 patients among 30 underwent R0 resection). Regarding patients enrolled prior to the trial’s closure (70 patients in the FFX-alone arm and 56 in the TNT arm), the primary endpoint (18-month OS rate) was 67.9% versus 47.3%, favoring the FFX-alone arm [95]. However, the particularly unfavorable oncological results obtained in the TNT arm are to be somewhat mitigated as the data published results from underpowered arms with highly heterogenous RT schemes performed with up to 12.5% of palliative RT treatment (25 Gy in five fractions) administered in the TNT arm. In fact, the delivery of a BED10 > 60 Gy appears to be associated with improved OS and PFS, as indicated by multivariate analysis [4,96]. As such, hypofractionated SBRT techniques were modified in order to reach a (nearly) ablative dose to improve efficacity.

##### Nearly Ablative SBRT

In a recent observational study, Simoni et al. evaluated the use of SBRT (delivered in five consecutive daily fractions by administering 30 Gy to the planning target volume while simultaneously delivering a 50 Gy SIB to the TVI), after induction FFX or gemcitabine/nab-paclitaxel chemotherapy (median 12 cycles) in 59 patients (54.2% LAPC and 45.8% BRPC) [102]. Among the 58 patients who completed SBRT, 59.4% were resected. No ≥G3 AEs were reported. Comparing resected and unresected patients, the one- and two-year freedom from local progression rates were 85% and 80% versus 79.7% and 60.6%, respectively (*p* = 0.33). The median OS was 30.2 months, and the median PFS was 19 months, for the entire cohort. Notably, resected patients had superior 2-year OS rates (72.5% versus 49%, *p* = 0.012) and median PFS (13 months versus 5 months; *p* < 0.001) compared to unresected patients. Additionally, no survival differences were observed between BRPC and LAPC [102].

An interesting approach to delivering high-BED SBRT, without compromising the safety on non-MR-Linac systems, involves employing a full isotoxic dose prescription (IDP), which relies on OAR tolerance thresholds rather than tumor volume, as conventionally practiced. By safeguarding the OARs, the isotoxic high dose (iHD)-SBRT technique enables the attainment of the maximum feasible dose level, enhancing the probability of local tumor control while ensuring a safe achievement of BED10 ≥ 70 Gy [26,111]. Bouchart et al. presented feasibility and initial efficacy findings of a TNT sequence, comprising preoperative FFX for six cycles (or gemcitabine/nab-paclitaxel if intolerant or unresponsive) followed by iHD-SBRT (SIB up to 53 Gy at the TVI in five fractions) [26]. Out of 39 patients (21 BR and 18 LAPC), 87.2% completed the neoadjuvant sequence, with 55.9% undergoing oncological resection after iHD-SBRT. Grade 3 early and late gastrointestinal toxicities were minimal (<5%). With a median follow-up of 18.2 months, median OS and PFS were 24.5 and 15.6 months, respectively. Resected patients had significantly better OS and PFS than non-resected patients, and there were also no survival differences between BRPC and LAPC patients. The 1-year local control rate from SBRT was 74.1%, and loco-regional PFS was not reached for either group [26]. However, these studies included both BRPC and LAPC patients, and efforts should be made to study these groups separately. Following those promising results, the randomized comparative phase II trial STEREOPAC (NCT05083247) aims to assess the efficacy of adding iHD-SBRT (both nearly and ablative SBRT allowed) to preoperative FXX in the neoadjuvant treatment of BRPC patients only (Table 3) [112].

##### Ablative SBRT

The increasing accessibility of magnetic resonance linear accelerator (MR-linac) systems in prominent radiation therapy centers presents an elegant solution for delivering high-dose radiation therapy [4]. Stereotactic magnetic resonance (MR)-guided adaptive radiation therapy (SMART) allows the administration of ablative doses by providing real-time visualization and delineation of gastrointestinal structures, thereby adjusting for anatomic variations [106]. This technique is backed by dosimetric studies and a prospective phase I clinical trial, demonstrating that MR-guided adaptive SBRT increased the target volume dosing while protecting critical OARs [106,113,114].

In 2023, a prospective phase II trial, led by Parikh et al., included 136 patients with BRPC (56.6%) and LAPC (43.4%) to study the use of induction chemotherapy with mFFX (65.4%) or gemcitabine/nab-paclitaxel (16.9%) (mean of 5.1 months) followed by SMART (50Gy in five fractions, BED10 = 100 Gy) [108]. The study achieved its primary objective, which was to show that the incidence of acute G3 or higher gastrointestinal toxicity directly attributed to SMART would be less than 15.8%. The occurrence of acute ≥ G3 gastrointestinal toxicity, potentially or probably associated with SMART, was 8.8%. This included two postoperative deaths that were possibly linked to SMART in patients who underwent surgery. The resection rate was 32.4%. One-year overall survival was 56% in non-resected patients and 85% in resected patients [108]. Recently, more mature results showed a 2-year OS of 53.6% and a median OS of 22.7 months [115]. Results from studies evaluating ablative SBRT are summarized in Table 2 [106,107].

## 5. The Advent of TNT for LAPC Patients

LAPC patients are also a population target of choice for the use of TNT strategies in order to increase the chance of surgical and oncological resection, R0 resection rates, and survival outcomes. Similarly to BRPC patients, the previously described TNT approaches have also been studied in LAPC, as well as those including HFA-IMRT, and are described below.

### 5.1. TNT with Conventional CRT with Induction or Consolidation Chemotherapy Prior to Surgery

There are only two phase III RCTs on induction chemotherapy followed by CRT in LAPC patients [60,65]. 

Firstly, the LAP07 trial by the Groupe Coopérateur Multidisciplinaire en Oncologie (GERCOR) aimed to evaluate the benefit of CRT after 4 months of induction chemotherapy with gemcitabine with or without erlotinib [60]. CRT significantly reduced local progression rates (32% versus 46%, *p* = 0.030) and showed a trend towards better median PFS (9.9 versus 8.4 months, *p* = 0.060) and longer median time without retreatment (6.1 versus 3.7 months, *p* = 0.020). However, the primary endpoint was not achieved (median OS: 15.2 for CRT versus 16.5 months for chemotherapy alone, *p* = 0.830). Regarding RT quality assurance, 44% of CRT-treated patients experienced minor and 18% had major RT protocol deviations, with major deviations correlating with worse survival (median survival: 13.4 versus 17 months in case of minor deviations, *p* = 0.055) and with higher G ≥ 3 toxicities [116]. Since the LAP07 trial was designed in 2005, it used a gemcitabine regimen, as multi-agent chemotherapy was not yet introduced. Despite decreased, loco-regional progression, the TNT arm had a higher rate of metastatic progression (60% versus 44%, *p* = 0.040), indicating that CRT might confer additional survival benefits, with the use of chemotherapies offering better distant disease control [4]. It is worth mentioning that the resection rate in the LAP07 trial was notably low (7% after chemotherapy alone and 3% after CRT) due to the definitive nature of these treatments, with surgical exploration considered only in rare cases with significant response [60]. 

More recently, Fietkau et al. published the first results of the CONKO-007 phase III RCT comparing chemotherapy alone (6 months of gemcitabine or FFX) versus TNT with induction chemotherapy (3 months) followed by CRT (50.4 Gy in 28 fractions) with concomitant gemcitabine in LAPC patients [65]. The primary endpoint was initially OS since the beginning of induction chemotherapy, but due to the delayed accrual of patients, it was changed to R0 resection rate resulting in a reduced sample size of 525 patients. After induction chemotherapy, 22% were excluded due to progression or toxicity leading to the randomization of 335 patients with 121 of them resected. Resection rates for randomized patients were similar between two arms (35.9% and 36.3%). This TNT approach resulted in an improvement in negative circumferential resection margin (CRM) R0 resection and pCR rates (0 versus 6%, *p* = 0.0013), although it did not significantly differ for the global R0 resection rate (18 versus 25% for TNT, *p* = 0.1433). R1 resections occurred significantly more in the chemotherapy-alone arm (*p* = 0.0085). Nevertheless, with a median follow-up of 16 months, this effect on resectability did not translate into significant PFS and OS differences (HR for median PFS of 0.919 [95% CI 0.702–1.203], *p* = 0.540, and HR for median OS of 0.964 [95% CI 0.760–1.225], *p* = 0.766), whereas the PFS rate tended to be higher in the CRT arm after 2 years (17.5 months +/− 0.04% vs. 24.1 months +/− 0.04%) [65].

Three additional phase II trials aimed to explore the same question with globally disappointing results and are summarized in Table 1 [61,63,64]. Furthermore, two meta-analyses examining randomized trials failed to reveal a statistically significant discrepancy in OS between patients undergoing neoadjuvant chemotherapy alone versus TNT with CRT, except in a subgroup analysis involving consolidation CRT after a minimum of 3 months of induction CT [117,118]. 

An alternative strategy of TNT with CRT consists in a regimen of induction CRT followed by consolidation or maintenance chemotherapy. A phase III RCT aimed to compare gemcitabine alone with induction CRT (60 Gy in 30 fractions) with concomitant 5-FU/cisplatin followed by maintenance chemotherapy with gemcitabine in LAPC patients [66]. Second intention surgery was authorized if patients had a good response to treatment. As such, five patients (4.2%) underwent surgery (two in the CRT arm and three in the gemcitabine-alone arm). Overall, median OS was 13 months in the gemcitabine-alone arm and 11.1 months in the TNT strategy [66]. 

Results of selected and representative studies can be found in Table 1 [62,80].

### 5.2. TNT with Hyprofractionnated Ablative Intensity-Modulated Radiotherapy (HFA-IMRT)

Using hypofractionated ablative radiotherapy (HFA-RT) after induction chemotherapy presents an interesting dose-escalation option in order to elevate the BED10 delivered to the tumor while still using a technique close to the well-known conventional CRT. Krishnan et al. investigated 200 LAPC patients who underwent induction chemotherapy followed by either conventional CRT (50.4 Gy in 28 fractions with concomitant chemotherapy) or HFA-IMRT (using an SIB technique in 15 to 28 fractions to obtain a BED10 > 70 Gy, with concurrent chemotherapy) [70]. Only 47 patients with tumors more than 1 cm from the nearest gastrointestinal mucosa could receive HFA-IMRT. The study found that HFA-IMRT resulted in better OS (median: 17.8 vs. 15 months, *p* = 0.030; 3-year OS: 31% vs. 9%) and local control (median local–regional recurrence-free survival: 10.2 vs. 6.2 months, *p* = 0.050). No increased toxicity was noted, and high BED delivery was the only predictor of improved OS in multivariate analysis [70]. A prospective study by the Sloan Kettering group examined 136 LAPC patients who received definitive HFA-IMRT (delivering BED ≥ 100 Gy; either 75 Gy in 25 fractions or 67.5 Gy in 15 fractions). After a median follow-up of 12 months, the median OS and freedom from local progression (FFLP) were not reached. Remarkably, the 2-year OS and FFLP rates were 55% and 78%, respectively, with a favorable toxicity profile [71]. Following the encouraging results from a prospective cohort study evaluating the association of HFA-RT in LAPC patients, Reyngold et al. presented in 2023 the results from the phase II MAIBE trial studying 3–6 months (median 3.5 months) of modified FFX or gemcitabine/nab-paclitaxel followed by hypofractionated HFA-IMRT (67.5 Gy in 15 fractions or 75 Gy in 25 fractions with concurrent capecitabine) in 47 LAPC [68,72]. The resection rate was 26% with 58.3% of R0 resection. Two-year OS was 38.9% (95% CI, 21.9–55.6%), including 37.1% (18.5–55.8%) in non-surgical and 39.4% (7.0–72.1%) in surgical groups. The 90 days post-surgery mortality was 0%, and nine surgical AEs were recorded in six patients [68]. However, the phase II RCT SCALOP from the Oxford group, aiming for 2 × 2 randomization between an induction with gemcitabine/nab-paclitaxel +/− nelfinavir followed by conventional CRT or HFA-IMRT (60 Gy in 30 fractions with concomitant capecitabine) did not show a significant survival benefit in favor of the TNT arm (median OS: 15.6 versus 16.9 months for TNT, *p* = NS) despite a good tolerance of the dose-escalated CRT [69]. Results from studies evaluating HFA-IMRT can be found in Table 1.

### 5.3. TNT with SBRT

The only analysis pooling findings from 19 SBRT trials involving LAPC patients revealed a median OS of 17 months (ranging from 5.7 to 47 months), alongside a 1-year OS rate of 51.6% and a 1-year local control rate (LCR) of 72.3% [119]. Unfortunately, considerable heterogeneity existed among these studies, the majority of which were small-scale retrospective series [88,91,94,100,101]. Moreover, the dosage and fractionation regimens varied significantly, leaving the establishment of an optimal TNT protocol with fractionated SBRT ambiguous [4]. Results from studies evaluating SBRT in LAPC patients are summarized in Table 2.

#### 5.3.1. Non-Ablative SBRT

In a meta-analysis conducted by Tchelebi et al., focusing on LAPC, SBRT (doses of ≥5 Gy per fraction) demonstrated a modest advantage in 2-year OS compared to conventional CRT (26.9 months versus 13.7 months, *p* = 0.004). However, differences in OS were not statistically significant at 1 year (53.7% versus 49.3%, *p* = 0.630) [120]. 

In 2015, a phase II trial evaluated a TNT approach consisting of three doses of gemcitabine followed by SBRT (33Gy in five fractions) in 49 LAPC patients [99]. Quality of life was assessed through the QLQ-PAN26 module before and at 4 weeks and 4 months after SBRT, revealing a notable improvement in pancreatic pain (*p* = 0.001) four weeks post-SBRT. The incidence rates of acute and late toxicities (primary endpoint), graded as ≥2, including gastritis, fistula, enteritis, or ulcer, were 2% and 11%, respectively. With a median follow-up of 13.9 months, median OS was 13.9 months, and freedom from local disease progression at one year was 78%. Four patients (8%) underwent margin-negative and lymph node-negative surgical resections [99]. Results from studies evaluating non-ablative SBRT in LAPC patients are summarized in Table 2 [100,101]. In light of these disappointing results, the delivery of higher BED10 to the tumor should be sought for LAPC, similarly to BRPC.

#### 5.3.2. Nearly Ablative SBRT

In a large retrospective study using a large national database of cT2-4N0-1M0 pancreatic cancer patients, 631 LAPC patients received SBRT (40 Gy/5 fractions) after induction chemotherapy, with a resection rate of 9.2% (84% R0), similar to the rates seen in the 7819 patients treated with CRT (50.4 Gy/28 fractions) [105]. SBRT, however, was associated with a superior OS in the multivariate analysis (HR 0.84, 95% CI 0.75–0.93, *p* < 0.001) [105]. 

Those encouraging results were followed by two phase II trials studying SBRT following chemotherapy in LAPC patients [103,104]. Comito et al. analyzed the use of SBRT (45 Gy in six fractions) preceded by a gemcitabine-based chemotherapy in 71% of the 45 LAPC patients [104]. Median PFS and median OS were 8 months and 13 months, respectively [104]. In the LAPC-1 phase II trial, 50 LAPC patients were treated with eight cycles of induction FOLFIRINOX, followed by SBRT (40 Gy in five fractions) [103]. During FOLFIRINOX treatment, 30 G3 or 4 AEs were observed, and following SBRT, two (5%) G3 or four AEs were noted. Additionally, two (5%) patients experienced gastrointestinal bleeding-related deaths within three months post-SBRT. Thirty-nine (78%) patients received the SBRT treatment. The resection rate was 12% with a 100% R0 resection rate and 33% pCR. The 1-year OS and PFS were 64% (95% CI: 50–76%) and 34% (95% CI: 22–48%), respectively. Median OS was 17 months in patients receiving SBRT and 7 months in patients who did not receive SBRT [103]. Studies using nearly ablative SBRT after induction chemotherapy are summarized in Table 2.

Results from the Belgian ongoing phase II TORPEDO trial (B3002023000168) evaluating the role of SBRT (40 Gy in five fractions) in BRPC and LAPC patients are highly awaited (Table 3).

#### 5.3.3. Ablative Adaptive SBRT

A multi-institutional retrospective study suggested that dose-escalated SMART might offer a survival benefit for patients with pancreatic cancer compared to more conventional MR-guided radiation treatments. In this study by Rudra et al., 44 patients with BRPC (23%) and LAPC (73%) received different radiation therapy regimens, including high-dose SBRT. The high-dose SBRT group was divided into two subgroups based on the BED: BED ≤ 70 Gy (40–55 Gy in 25–28 fractions or 30–35 Gy in five fractions) and BED > 70 Gy group (50–67.5 Gy in 10–15 fractions or 40–52 Gy in five fractions). The 2-year OS rate was 49% in the high-dose group versus 30% in the low-dose group (*p* = 0.030). However, it is noteworthy that only six patients had surgery post-RT, and high-dose RT was not an independent predictor of OS in multivariate analysis [109]. Additional results from other key retrospective studies using this technique are summarized in Table 2 [106,107,110]. 

Although the US LAP-ABLATE trial evaluating MRI-guided ablative SBRT in LAPC patients will not open, the results from LAPSTAR (NCT06272162) are highly anticipated (Table 3).

## 6. Evaluation of Treatment Response and Future Perspectives for Patient Selection in the Precision Medicine Era

### 6.1. Available Tools for the Evaluation of Treatment Response

During TNT, a proper evaluation of treatment response, usually performed through clinical, biochemical, radiologic, and/or metabolic features, is of outmost importance, considering the length of the treatment period and the possibility of therapy switching [42]. 

**Carbohydrate antigen (CA) 19-9** stands as the only biomarker endorsed by the US FDA for monitoring therapy response in PDAC, playing a crucial role in clinical staging and treatment strategies for both localized and metastatic scenarios. Nevertheless, its utility is constrained to patients with a sialyl-Lewis A-positive genotype, encompassing roughly 90% of patients. Moreover, accurate assessment of CA 19-9 readings necessitates normal bilirubin levels, given that elevated CA 19-9 levels can also signify underlying inflammatory conditions [121]. For instance, in a retrospective series involving 131 patients initially presenting elevated CA19-9 (>35 U/dl) who underwent preoperative therapy and subsequent resection, the normalization of CA19-9 emerged as a robust prognostic marker for long-term survival rather than the magnitude of its reduction [122]. Another small study underscored the unlikelihood of achieving a major pathologic response in patients with persistently elevated CA19-9 after preoperative therapy. Notably, among 28 patients demonstrating a major pathologic response, 27 (96%) showed post-treatment CA19-9 within the normal range, despite 75% of them having elevated CA19-9 levels before treatment [123]. Furthermore, a recent systematic review encompassing 17 pertinent studies with comprehensive data on CA19-9 response during neoadjuvant therapy emphasized that achieving a post-neoadjuvant CA19-9 response exceeding 50% or achieving normalization was associated with improved overall survival [124]. 

While a consensus on resectability criteria at diagnosis remains elusive, the same challenge persists post-neoadjuvant treatment, with a specific hurdle emerging after TNT due to the fibrotic alterations induced by radiotherapy. 

**Multi-phase contrast-enhanced computed tomography (CT)** currently stands as the most validated tool for staging PDAC. Adding a layer of intricacy, response to neoadjuvant therapy might not always be reflected by radiographic indicators [125]. In the neoadjuvant setting, preoperative CT re-staging may lack specificity in distinguishing residual viable tumors from post-treatment-induced changes (with no viable tumor) particularly near involved vessels, due to the dense fibrous stroma from pancreatic tumors. In addition, after chemoradiotherapy, cancer cells decrease or disappear leaving behind fibrotic tissue that cannot be distinguished from tumor tissue. Some studies suggest that preoperative therapy could lead to radiographic downstaging in unresectable PDAC, potentially improving resectability, akin to its effects like breast and rectal cancers. Yet, these findings are hindered by inconsistencies in therapeutic response metrics, resection criteria, surgical techniques, and staging definitions [125,126]. Katz et al. retrospectively studied 129 BR PDAC patients to assess how NAT affected tumor size reduction or stage of BR tumors, based on standardized criteria for the indication of resection (AHPBA and MD Anderson) and clinical and pathological response criteria (e.g., using RECIST criteria) [21,127,128]. Their findings showed minimal clinical downstaging according to AHPBA criteria (1%), with 19% showing disease progression, and 80% demonstrating no change. Results were even less favorable with MD Anderson criteria, with 1% showing downstaging, 21% progressing, and 78% remaining unchanged. However, intraoperative findings revealed a more positive chemotherapy response, as 66% of patients underwent resection, with 95% achieving R0 resection [125]. Histopathologic analysis of pancreatectomy specimens further revealed favorable grade III or IV responses in 17% of resected patients, suggesting a promising prognosis despite persistent anatomical relationships in CT scans [125,129]. Interestingly, while radiographic changes in primary tumor anatomy were rare, histopathological analysis indicated significant cytotoxic activity at the tumor–vessel interface, with 74% of patients showing >1 mm distance from the primary tumor to the inked superior mesenteric artery (SMA) margin in cases with negative SMA margins [125]. Notably, RECIST response did not correlate with OS [125]. Perri et al. examined responses to first-line chemotherapy (FOLFIRINOX or gemcitabine/nab-paclitaxel) in 485 treatment-naive patients with localized pancreatic cancer, finding higher rates of RECIST partial responses with FOLFIRINOX compared to gemcitabine/nab-paclitaxel (19% vs. 6%; *p* = 0.001) but no significant differences in median tumor volume change, local tumor downstaging rates, or CA19-9 levels (35). Similarly, in a recent study of 194 patients undergoing TNT, anatomical downstaging was uncommon (28%) and not linked to survival, but 94% achieved negative margins with R0 resection, while radiologic anatomical downstaging was uncommon (28%) and not correlated with survival, and 94% of patients achieved negative margins (R0 resection) [7]. Only three factors were independently associated with prolonged survival, including extended duration (≥6 cycles) chemotherapy, optimal post-chemotherapy CA19-9 response, and major pathologic response. Moreover, among 67 patients who underwent interval metabolic positron emission tomography (PET) imaging post-chemotherapy, complete metabolic response strongly correlated with major pathologic response [7].

### 6.2. Future Perspectives for Response Status and Patient Selection

Alternative imaging techniques, such as magnetic resonance (MR) imaging with specific sequences, show promising potential and are currently being studied in conjunction with metabolic imaging and conventional CT [130]. Although **18F-FDG PET** is not standard in clinical practice, it was proven to be sensitive for initial TNM staging, treatment response, and recurrence detection [131,132,133]. Furthermore, 18F-FDG PET/CT imaging may predict treatment efficacy and clinical outcome for PDAC [134]. Some centers currently rely on post-chemotherapy metabolic responses, evaluated through PET/CT or PET/MRI as surrogate markers for treatment response assessment [135,136]. Reductions in SUVmax, SUVpeak, and MTV have been associated with improved overall survival [7,135,137]. PET/CT or PET/MRI-based metabolic response may offer greater sensitivity in assessing tumor response compared to traditional CT or MRI and warrants further investigation in clinical trials [137]. However, 18F-FDG PET/CT has limitations in PDAC due to variable detection of metastatic lymph nodes and potential false-positive results in cases of inflammation [138,139]. Additionally, promising results have emerged from the use of radiomics-based machine-learning models in diagnosing pancreatic cancer, demonstrating their potential to predict diagnoses earlier than human observers [140]. Radiomics might assist in patient selection for TNT by analyzing quantitative imaging features to predict individual responses to treatment. By assessing tumor characteristics, such as texture, heterogeneity, and vascularity, radiomics can identify patients likely to benefit from TNT, guiding personalized treatment decisions. Additionally, radiomics can help monitor treatment response over time, allowing for early adaptation of therapy if necessary. Ultimately, by providing insights into tumor biology and treatment response, radiomics enables more tailored and effective TNT strategies for pancreatic cancer patients.

Compared to other solid tumors, the tumor microenvironment (TME) of PDAC is uniquely complex and dynamic due to intensive interactions among its components. PDAC is characterized by intense stromal desmoplastic reactions surrounding cancer cells, creating an acidic and hypoxic microenvironment, with high interstitial pressure. This environment acts as a sanctuary, limiting drug penetration and impeding immune cells recruitment [141,142,143]. Cancer-associated fibroblasts (CAFs) are key players in the desmoplastic reaction, comprising up to 90% of the tumor tissue in carcinomas. Fibroblast activation protein (FAP), a type-II transmembrane serine protease, is prominently expressed in CAFs [144]. Given its widespread expression, FAP is an attractive target for radionuclide imaging with PET/CT [141,143,145,146]. FAP-targeted imaging has shown promising diagnostic sensitivity and specificity, with a low tracer uptake in non-tumoral tissues across various cancers, including PDAC [147]. Consequently, pancreatic cancer is expected to show intensive uptake of 68Ga-FAPI, encouraged by previous studies [148,149]. Biologically, 68Ga-FAPI-PET in PDAC is of special interest as it depicts tumor–stroma interaction, crucial for the tumorigenesis of PDAC, aspects not captured by morphologic or metabolic imaging modalities. Recently, Koerber et al. underscored the clinical impact of 68Ga-FAPI PET/CT imaging in 19 patients with PDAC (seven primary and 12 progressive/recurrent), highlighting its ability to detect new lesions or clarify ambiguous findings from standard CT scans. The group from Heidelberg found that biodistribution analyses revealed high FAPI uptake in primary PDAC, lymph nodes, and distant metastases, while healthy tissues exhibited minimal background activity, resulting in excellent tumor-to-background ratios [147,149,150]. The Heidelberg group also noted discrepancies between 18F-FDG PET and 68Ga-FAPI PET-based staging, often leading to upstaging with the latter, consistent with findings in other tumor types [150]. As such, 68Ga-FAPI PET may lead to significant changes in treatment decisions for PDAC when compared with 18F-FDG PET and is a key innovant player in improving PDAC management. A study addressing this specific question in managing PDAC patients treated with TNT is currently ongoing at our center.

Although some patients achieve pCR, more than half relapse after surgery [151].

**Liquid biopsies** enable the collection of repeated samples during neoadjuvant therapy and can help to monitor treatment response and to stratify patients to the most suitable treatment [31,152]. Notably, Yin et al. discovered the persistence of somatic mutations, circulating tumor cells (CTCs), and circulating tumor DNA (ctDNA) in individuals with pancreatic cancer who achieved pCR following neoadjuvant therapy. CTCs were found in five of six patients and ctDNA in 7 of 16 patients with pCR. This led to the introduction of a novel concept of molecular complete response (mCR), by amalgamating genomic analysis of resected specimens with liquid biopsy data [153]. In a study of 59 patients, preoperative ctDNA was detected in 69% of those directly undergoing surgery compared to 21% in patients receiving neoadjuvant treatment. The presence of preoperative ctDNA correlated with significantly lower recurrence-free survival (RFS) and overall survival (OS). Notably, patients with persistent ctDNA after neoadjuvant treatment all experienced relapse, with a median RFS of 5 months [154]. Monitoring KRAS mutant allele frequency (MAF) in exoDNA during NAT may also help with predicting the resectability of PDAC. Indeed, a study revealed that 71% of patients eligible for PDAC resection exhibited a decreased KRAS MAF, whereas in 94% of patients with unresectable PDAC post-NAT, KRAS MAF remained stable or increased [155]. Furthermore, Bernard et al. discovered that an increase in exoDNA after NAT was significantly associated with disease progression, suggesting that monitoring exoDNA during NAT using liquid biopsy provides predictive insight and prognostic information relevant to therapeutic stratification [156].

While personalized medicine in pancreatic cancer has made strides, it encounters persistent barriers hindering effective clinical application [30]. Presently, patients with BR and LAPC adhere to a standardized treatment strategy drawn from RCTs on neoadjuvant therapy, maintaining a one-size-fits-all paradigm [157,158,159,160]. However, substantial variability exists in survival and response to neoadjuvant therapy with FOLFIRINOX and gemcitabine/nab-paclitaxel. Despite FOLFIRINOX demonstrating higher RECIST partial response rates and subsequent pancreatectomy, overall survival remains comparable between regimens [161,162]. The absence of biomarkers for chemotherapy selection necessitates a transition toward more effective and tolerable regimens, guided by enhanced patient stratification. A notable breakthrough involves a patient-derived organoid library revealing gene expression signatures predictive of improved chemotherapy response [163]. This innovative approach holds promise for identifying genomic, transcriptomic, and therapeutic profiles, paving the way for tailored treatments that swiftly confer clinical benefits to each patient.

Pancreatic cancer patients may benefit from **tumor molecular profiling** and targeted therapies [164]. Transcriptomic signatures, such as GemPred, have been studied as predictive tools of response to adjuvant chemotherapy in PDAC [165]. Likewise, PAMG has been associated to progression under mFFX treatment (*p* < 0.001) [166]. Similarly, other gene expression signatures used to predict response to neoadjuvant CRT in PDAC patients have been investigated [167]. Predictive molecular signatures are required to forecast the response to TNT, enabling the molecular selection of patients. A retrospective analysis of the Know Your Tumor registry trial revealed that patients with actionable molecular alterations had improved outcomes with matched therapies [168]. However, only a small percentage (about 8%) of pancreatic cancer patients harbor potentially targetable genetic alterations [168]. Current treatment recommendations mainly apply to metastatic cases, but exploring these findings in earlier stages is warranted in future trials. ESMO suggests multi-gene sequencing for advanced pancreatic cancer patients in molecular screening programs, enabling access to an innovative drug [169]. Currently, there is limited evidence for specific neoadjuvant regimens other than FOLFIRINOX and gemcitabine/nab-paclitaxel. In the future, the utilization of molecular and mutational signatures could guide treatment decisions, influencing the selection between gemcitabine-based regimens, FOLFIRINOX, or radiotherapy.

## 7. Conclusions

The advent of TNT presents a promising avenue for both BRPC and LAPC patients, aiming to optimize outcomes through tailored radiotherapy and chemotherapy regimens. The Dutch PREOPANC regimen has shown superior R0 resection rates, highlighting the efficacy of integrating preoperative gemcitabine with hypofractionated CRT. Ongoing research, including the exploration of SBRT, offers potential for precise tumor targeting and improved survival rates. Ablative SBRT techniques, especially when combined with induction chemotherapy, demonstrate promise in enhancing overall survival and facilitating oncological resection. These innovative approaches underscore the evolving landscape of neoadjuvant therapy, offering hope for improved outcomes in pancreatic cancer management.

Current perioperative trials prioritize the assessment of optimal treatment regimens and sequences, often employing gemcitabine-abraxane or FOLFIRINOX for a minimum duration of 4 months with possibilities of switching and with a growing focus on the integration of SBRT, which offers the advantage of a very short course administration. Refining therapeutic protocols, especially by extending the duration of chemotherapy prior to surgery and incorporating refined SBRT techniques, is expected to lead to substantial advancements in pancreatic cancer management. The TNT strategy holds great potential, and conok the most favorable therapeutic outcomes.

## Figures and Tables

**Table 1 cancers-16-02423-t001:** Main studies evaluating TNT in BRPC and LAPC.

Study	Study Design	Type	N	Treatment		Resection (%)		Survival/Progression		
				Neoadjuvant Treatment	Adj Cht (%)	R Rate	R0 Rate	DFS	PFS	OS
Induction Cht → CRT or CRT → Consolidation Cht										
Dholakia 2013 [57]	Retrosp	BRPC	50	mFFX/Gem-based → CRT50 Gy/25 (Cap/Gem)	NA	58	93	NA	13.4 M **NR/R: 16.7/5.9 M** **(*p* < 0.001) ***	17.2 M **NR/R: 22.9/13 M** **(*p* < 0.001) ***
Katz 2016 [13] (Alliance A021101)	Phase II	BRPC	22	mFFX (4c) → CRT 50.4 Gy/28 (Cap)	45	68	93	NA	NA	21.7 M
Nagakawa 2017 [56]	Phase II	BRPC	27	Gem (2c) → CRT 50.4 Gy/28 (Gem+S-1)	94.7	70	94.7	NA	NA	22.4 M 1 y OS 81.3% 2 y OS 33.9%
Wo 2018 [59]	Retrosp	BRPC LAPC	99	FFX /FOLFOX/Gem based (4-8c) → CRT 58.8 Gy/28 (Cap/5FU/Gem)	NA	37	87	NA	NA	18.1 M
Pietrasz 2019 [6] AGEO-FRENCH	Retrosp	BRPC LAPC	203	FFX (6c)	73.2	73.2	76.3	13.5 M	NA	35.5 M
				FFX (6c) → CRT 54 Gy/30 (5-FU or Cap)	41.2	**41.2**	**89.2** **(*p* = 0.017) ***	17.7 M (*p* = 0.121)	NA	**57.8 M** **(*p* = 0.007) ***
Auclin 2021 [58] AGEO cohort	Retrosp	BRPC LAPC	330	FFX (7c) → CRT 50.4 Gy/28 (Cap)	NA	13.8	74.7	12.8 M	NA	21.4 M
Hammel 2016 [60] LAP07	Phase III Rando	LAPC	449	Gem +/− Erlotinib 4c → 2c CRT	NA	6	2.5	NA	NA	16.5 M
				Gem +/− Erlotinib 4c → CRT 54 Gy/30	NA	3		NA	NA	15.2 M
Sudo 2017 [61]	Phase II	LAPC	30	S1+Gem → CRT 50.4 Gy/28 (S1)	70	10	100	NA	12.7 M	21.3 M
Huguet 2017 [62]	Retrosp	LAPC	134	Gem-based/FFX → CRT 50.4 Gy/28 (Gem/Cap)	NA	19	85	NA	NA	2 M 1 y OS 86% 2 y OS 48%
Sherman 2018 [63]	Phase II	LAPC	45	GTX → CRT 56 Gy/28 (Cap)	NA	89	70	R0 31 M R1 14 M	NA	R0: 37 M R1: 29 M
Loka 2021 [64]	Phase II Rando	LAPC	102	S1 → CRT 50.4 Gy/28 (Gem)	NA	NA	NA	NA	NA	1 y OS 66.7% 2 y OS 36.9%
				Gem 3c → CRT 50.4 Gy/28 (Gem)	NA	NA	NA	NA	NA	1 y O 69.3% 2 y OS 18.9%
Fietkau 2022 [65] CONKO-007	Phase III Rando	LAPC	335	Gem 6c/FFX (12c)	NA	35.9	18	NA	1 yOS 59.0% 2 yOS 17.5%	1 y OS 71.3% 2 y OS 32.5%
				Gem (3c) /FFX (6c) → CRT 50.4 Gy/28 (Gem)	NA	36.3	25	NA	1 y OS 56.3% 2 y OS 24.1%	1 y OS 71.1% 2 y OS 34.8%
Barhoumi 2011 [66]	Phase III	LAPC	119	CRT 60 Gy/30 (5FU+CDDP) → Gem	NA	NA	NA	NA	NA	8.6 M 1 y OS 53%
				Cht (Gem)						**13 M (*p* = 0.03) *** 1 y OS%
PREOPANC										
Versteijne 2022 [67] PREOPANC1	Phase III Rando	RPC BRPC	246	Immediate surgery	51	72	43	7.7 M	LFFI 13.4 M	14.3 M
				CRT: 36 Gy/15 (Gem)	46	61	72 (*p* < 0.001)	8.1 M	LFFI 31.2 M (*p* = 0.04)	15.7 M
Koerkamp 2023 [41] PREOPANC2	Phase III Rando	RPC BRPC	375	FFX (8c)	NA	77	NA	NA	NA	21.9 M
				CRT: 36 Gy/15 (Gem)	Yes	75	NA	NA	NA	21.3 M
HFA-IMRT										
Reyngold 2023 [68]	Prospec	BRPC	47	mFFX/Gem-based → CRT 75 Gy/25; 67.5/15 (Cap)	NA	26	58.3	NA	NA	2 y OS 38.9% 2 y OS (NR/R): 37.1%/39.4%
Mukherjee 2022 [69] SCALOP-2	Phase II	LAPC	186	Gem/Nab-P+/−Nelfinavir → CRT 50.4 Gy/28 (Cap)	NA	NA	NA	1 yPFS 33.3%	1 yLRF 26.7%	15.6 M
				Gem/Nab-P+/−Nelfinavir → HFA 60 Gy/30				1 yPFS 23.9%	1 yLRF 15.2%	16.9 M
Krishnan 2016 [70]	Retrosp	LAPC	200	FFX/Gem-based → CRT 50.4 Gy/28 (Cap/Gem)	NA	NA	NA	NA	LR RFS 6.2 M	15 M 3 y OS 9%
				FFX/Gem-based → HFA (BED > 70 Gy)					LR RFS 10.2 M	17.8 M 3 y OS 31%
Reyngold 2019 [71]	Prospec	LAPC	136	CMT → RT 75 Gy/25; 67.5 Gy/15; SBRT 50 Gy/5	NA	NA	NA	NA	2 y FFLP 78%	2 y OS 55%
Reyngold 2021 [72]	Prospec	LAPC	119	mFOLFIRINOX/Gem Based → RT 75 Gy/25; 67.5/15	NA	NA	NA	NA	6.3 M 1 yLRF 17.6% 2 yLRF 32.8%	18.2 M

Abbreviations: * statistically significant; 5-FU: 5-fluoracil; Adj: adjuvant; BRPC: borderline resectable pancreatic cancer; c: cycle; Cap: capecitabine; CDDP: cisplatin; Cht: chemotherapy; CRT: combined chemoradiotherapy; DFS: disease-free survival; FFLP: freedom from local progression; FFX: FOLFIRINOX; Gem: gemcitabine; GTX: gemcitabine, docetaxel, and capecitabine; HFA-IMRT: hypofractionated ablative intensity-modulated radiotherapy; LAPC: locally advanced pancreatic cancer; LFFI: loco-regional failure-free interval; LR: local–regional; LRF: local–regional failure; M: months; N: number; NA: not available; NR: non-resected; OS: overall survival; Prospec: prospective; PFS: progression-free survival; R: resected; Rand: randomized; RCP: resectable pancreatic cancer; RFS: recurrence-free survival; RT: radiotherapy; Retrosp: retrospective; TNT: total neoadjuvant treatment; Y: year.

**Table 2 cancers-16-02423-t002:** Main studies evaluating SBRT in BRPC and LAPC.

Study	Study Design	Type	N	Treatment		Resection (%)		Survival/Progression		
				Neoadjuvant Treatment	Adj Cht (%)	R Rate	R0 Rate	DFS/PFS	LC	OS
Non-Ablative Stereo (BED10 [50–60 Gy])										
Chuong 2016 [97]	Retrosp	BRPC	36	GTX → SBRT 35 Gy/5 (32.5–40 Gy)	No	88.6	97.2	mPFS:14.9 M	NA	22.5 M
Chuong 2013 [98]	Retrosp	BRPC LAPC	73	Gem (3c) → SBRT 25–30 Gy/5 (SIB TVI 35–50 Gy)	NA	BRPC: 56	BRPC: 97	mPFS BRPC 9.7 M LAPC 9.8 M 1 yPFS BRPC: 42.8% LAPC: 41%	1 y LC (NR): 81%	BRPC 16.4 M LAPC 15 M 1 y OS BRPC: 72.2% LAPC: 68.1%
Mellon 2015 [87]	Retrosp	BRPC LAPC	159	Gem/mFFX → SBRT 28–30/5 (SIB TVI 50)	NA	BRPC: 51 LAPC: 14	BRPC: 96 LAPC: 100	mPFS BRPC 19.2 M LAPC 15 M	NA	BRPC 19.2 M LAPC 15 M
Moningi 2015 [88]	Retrosp	BRPC LAPC	88	Gem/mFFX → SBRT 25–33 Gy/5	NA	22	84	mPFS 9.9 M 1 y PFS 41% 2 y PFS 11%	13.9 M 1 y LC: 61% 2 y LC: 14%	13.7 M 1 y OS 60% 2 y OS 15%
Mellon 2017 [89]	Retrosp	BRPC LAPC	222	Gem/FFX (3c) → SBRT 25–30/5 (SIB TVI 50 Gy)	NA	BRPC: 51 LAPC: 11	97.5	NA	NA	37.5 M
Quan 2018 [90]	Phase II	BRPC LAPC	35	Gem/Cap 4c → SBRT 36 Gy/3	NA	BRPC: 53 LAPC: 12.5	91.7	NA	1 y LC (R/NR) 80/44%	BRPC 18.8 M LAPC 28.3 M
Gurka 2017 [91]	Retrosp	BRPC LAPC	38	Concurrent Gem based/mFOLFOX → SBRT 25–30 Gy/5	NA	NA	NA	mPFS 6.8 M	6 M LC 82%	12.3 M
Kharofa 2019 [92]	Phase II Rando	BRPC RPC	18	Gem or FFX 3c → SBRT 25 Gy/5 (SIB TVI 33 Gy)	NA	67	92	mPFS 1 M	1 y LC 50%	21 M
Palta 2018 [93]	Phase II	BRPC RPC	25	Gem/Nab-P (2c) → SBRT 25 Gy/5	NA	68	93	NA	1 y LC 77%	24 M
Jiang 2019 [94]	Retrosp	BRPC LAPC	5828	CMT + SBRT (5.7%)	NA	NA	84.9	NA	NA	32.1 M
				CMT (55.5%)			75.9			27.5 M
				CMT + CRT (38.8)			83.2			27.1 M
Katz 2021 [95] Alliance A021501	Phase II Rando	BRPC LAPC	126	mFFX 8c	0.34	58	88	NA	NA	29.8 M 18 M-OS 66.7%
				mFFX (7c) → SBRT 25–33 Gy/5 (SIB TVI 40 Gy)	0.24	51	74			17.1 M 18 M-OS 47.3%
Herman 2015 [99]	Phase II	LAPC	49	Gem → SBRT 33 Gy/5	Yes	8	100	mPFS 7.9 M 1 y PFS 32% 2 y PFS 10%	1 y LC 78%	13.9 M 1 y OS 59% 2 y OS 18%
Jumeau 2018 [100]	Retrosp	LAPC	17	Gem/FFX (38%) → SBRT 30 Gy/5	NA	NA	NA	NA	6 M LC: 93%1 y LC 67%	22 M
Jung 2019 [101]	Retrosp	LAPC	95	Gem Based/mFFX → SBRT 24–36 (median 28 Gy/4)	NA	7.4	57	mPFS 10.2 M 1 y PFS 42.9%	NA	16.7 M 1 y OS 67.4%
Nearly Ablative Stereo (BED10 [70–85 Gy])										
Bouchart 2021 [26]	Obs	BRPC LAPC	39	mFFX (6c) → SBRT 25–30/5 (SIB TVI 50 Gy)	NA	55.9	NA	mPFS 15.6 M **NR/R 7/24.1 M** ***p* < 0.001**	1 y LC 74.1% (SBRT)	24.6 M **NR/R: 18.2/32.3 M** ***p* = 0.02**
Simoni 2021 [102]	Obs	BRPC LAPC	59	Gem-based/FFX → SBRT 25–30/5 (SIB TVI 50 Gy)	NA	59.4	NA	mPFS 10.7 M **NR/R 5/13 M** ***p* < 0.001**	1 y LC (NR/R) 79.7/85% 2 y LC (NR/R) 60.6/80%	19.1 M **2 y OS (NR/R) 49/72.5%** ***p* = 0.012**
Suker 2019 [103] LAPC-1	Phase II	LAPC	50	FFX → SBRT 40 Gy/5	NA	12	100	mDFS SBRT 11 M non-SBRT 3 M mPFS 9 M 1 y PFS 34%	SBRT 20 M non-SBRT 3 M	SBRT 1 M non-SBRT 7 M 1 y OS: 64%
Comito 2017 [104]	Phase II	LAPC	45	Gem-based → SBRT 45 Gy/6	0.48	NA	NA	mPFS 8 M	mFFLP: 26 M 2 y OS 90%	13 M
Zhong 2017 [105]	Retrosp	LAPC	7819	CRT 50.4 Gy/28	NA	9.2	84	NA	NA	11.6 M 2 y OS 16.5%
			631	SBRT 40 Gy/5		10.8	92			13.9 M 2 y OS 21.7% (*p* = 0.014) *
Ablative Stereo (BED10 [100 Gy])										
Hassanzadeh 2021 [106]	Retrosp	BRPC LAPC	44	FFX or Gem/Nab-paclitaxel or Gem alone → SBRT 50 Gy/5	43.2	9.1	50	mDFS 21.3 M 1 y DFS 78% 2 y DFS 37.8% mPFS 12.4 M 1 y PFS 52.3% 2 y PFS 13.9%	1 y LC: 84.3% 2 y LC: 59.3%	15.7 M 1 y OS 68.2% 2 y OS 37.9%
Chuong 2022 [107]	Retrosp	BRPC LAPC	62	FFXor Gem/Nab-P → SBRT 50 Gy/5	NA	22.6	NA	mPFS 20 M 1 y PFS 88.4% 2 y PFS 40%	1 y LC 98.3% 2 y LC 87.7%	23 M 1 y OS 90.2% 2 y OS 45.5%
Parikh 2023 [108]	Phase II	BRPC LAPC	136	FFX or Gem/Nab-P → SBRT 50 Gy/5	19.9	32.4	NA	1 y PFS (NR/R): 46%/82%	1 y LC (NR/R): 78%/93	1 y OS (NR/R): 56%/85%
Rudra 2019 [109]	Retrosp	LAPC	44	BED < 70 Gy: CRT (40–45 Gy/25–28)/SBRT (30–35 Gy/5)	NA	NA	NA	18 M FFDF 48%	2 y FFLF 57%	2 y OS 30%
				BED > 70 Gy: CRT (50–67.5 Gy/10–15)/SBRT (40–52 Gy/5)				18 M FFDF 24%	2 y FFLF 77%	**2 y OS 49%** **(*p* = 0.03)**
Tringale 2022 [110]	Retrosp	LAPC	30	FFX → SBRT 50 Gy/5	NA	NA	NA	mPFS 10.1 M 1 y PFS 39.7%	NA	1 y OS 96.4%

Abbreviations: * statistically significant; 5-FU: 5-fluoracil; Adj: adjuvant; BED: biologically effective dose; BRPC: borderline resectable pancreatic cancer; c: cycle; Cap: capecitabine; Cht: chemotherapy; CRT: combined chemoradiotherapy; DFS: disease-free survival; FFDF: freedom from distant failure; FFLF: freedom from local failure; FFLP: freedom from local progression; FFX: FOLFIRINOX; Gem: gemcitabine; GTX: gemcitabine, docetaxel and capecitabine; LAPC: locally advanced pancreatic cancer; LC: local–regional control; m: median; M: months; N: number; NA: not available; NR: non-resected; OS: overall survival; Prospec: prospective; PFS: progression-free survival; R: resected; Rand: randomized; RCP: resectable pancreatic cancer; RFS: recurrence-free survival; RT: radiotherapy; Retrosp: retrospective; SBRT: stereotactic body radiation therapy; Y: year.

**Table 3 cancers-16-02423-t003:** Main ongoing studies evaluating TNT in PDAC.

Study NCT	Study Design	Recruitment Status	Type	N	Treatment		Primary Endpoint
					Study Treatment	Ajd Cht	
**Bouchart STEREOPAC** NCT05083247	Phase II Random	Recruiting	BRPC	256	8c mFFX/12c Gem/Nab-P	NA	R0 DFS
					6c mFFX or 9c Gem/Nab-P → SBRT 35–55 Gy/5	NA	
**AGITG Masterplan** ACTRN12619000409178	Phase II Random	Recruiting	RPC BRPC LAPC	120	6c mFFX	Yes	LC 1 year
					6c mFFX → SBRT 40 Gy/5		
**PANDAS PRODIGE 44** NCT02676349	Phase II Random	Active Not recruiting	BRPC	130	mFFX	Yes	R0 rate
					mFFX → CRT 50.4 Gy/28(Cap)		
**Span-C trial** NCT03505229	Phase II	Recruiting	BRPC LAPC	40	Oxaliplatin-based regimen/Gem-based regimen → SBRT 30–45 Gy/5	NA	Local failure at 12 M
**Wei** NCT05851924	Phase II	Recruiting	BRPC LAPC	60	FFX →AD-RT	NA	EFS
**TORPEDO** **B3002023000168**	Phase II	Starting	BRPC LAPC	NA	FFX (6c) or Gem-Nab-P (3c)	NA	NA
					FXX (2c) + SBRT 40 Gy/5		

Abbreviations: ablative dose radiotherapy; Adj: adjuvant; BRPC: borderline resectable pancreatic cancer; c: Cycle; Cht: chemotherapy; DFS: disease-free survival; FFX: FOLFIRINOX; Gem: gemcitabine; LC: loco-regional control; N: number; NA: not available; OS: overall survival; R0: R0 resection rate; Random: randomized; SBRT: stereotactic body radiation therapy.

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
