# Peer review of "Total Neoadjuvant Therapy in Localized Pancreatic Cancer: Is More Better?"

_cancers, 2024, doi:10.3390/cancers16132423_

Round 1
Reviewer 1 Report
Comments and Suggestions for Authors
This is a well written and comprehensive review of the field that should be informative to readers. My only concern is that there is so much data that it is hard to find a consistent message. Perhaps a summary table might provide a way to get important points across.
Author Response
We appreciate your feedback on our review. We aimed to provide a comprehensive overview without overwhelming the readers, which is why we decided against adding a fourth table. Instead, we focused on summarizing the information more concisely within the existing three tables.
Reviewer 2 Report
Comments and Suggestions for Authors
In this review entitled “Total neoadjuvant therapy in localized pancreatic cancer: is more better?” by Saúde-Conde Rita et al discussed the total neoadjuvant therapy (TNT) on improving borderline resectable (BRPC) and locally advanced (LAPC) pancreatic cancer patients treatment outcome with a focus on surgical success and overall survival. The authors addressed a critical question in the field: would more neoadjuvant therapy in a pre-surgery setting always be beneficial to patients? They categorized TNT strategies into 6 sub-groups and collectively layout clinical trial evidence in each group in great detail to discuss and compare the advantages and disadvantages of different TNT strategies. In addition, the authors discussed useful imaging tools or molecular markers to evaluate the treatment response during TNT and its potential application in patient selection for precision medicine. The authors have concluded that TNT indeed provides benefits to both BRPC and LAPC patients and with better strategies would yield better outcomes. The review is well-written and comprehensively investigates each topic and focuses on important parameters to make logical conclusions. I only have minor concerns listed below:
Line 129: “Furthermore, the 90-day mortality was lower in both groups.” It is confusing to me which group to compare.
Line 308, 371, and 404 have shown multiple abbreviations of “organs at risk (OARs)”.
Line 632, is the first time mentioning the superior mesenteric artery (SMA) margin, please include the whole phrase.
Author Response
We appreciate your feedback on our review.
Comment 1: Line 129: “Furthermore, the 90-day mortality was lower in both groups.” It is confusing to me which group to compare.
Response 1: Thank you for your comment. The sentence was corrected for a better understanding.
Comment 2: Line 308, 371, and 404 have shown multiple abbreviations of “organs at risk (OARs)”.
Response 2: Thank you for pointing this out. Corrections have been made.
Comment 3: Line 632, is the first time mentioning the superior mesenteric artery (SMA) margin, please include the whole phrase.
Reponse 3: Thank you for noticing this, corrections have been made.
Reviewer 3 Report
Comments and Suggestions for Authors
This is a very comprehensive work on the current status of systemic chemotherapy alone and in combination with radiation therapy for pancreatic cancer.It shows how the best therapeutic approaches have been and are being developed based on many successive studies.Stereotactic body irradiation in combination with induction chemotherapy is particularly important with regard to survival rates.The reference to the excellent possibility of post-therapeutic follow-up with liquid biopsy and circulating tumor DNA,which is also used in follow-up studies of other tumor therapies,is also very essential and well presented. A list of the side-effects that undoubtedly severely limit the quality is largely missing,but it does not belong to the present topic of possible therapy options and their possible optimization and can therefore be omitted and published under other headings.From this perspective the same applies to intra-arterial chemotherapy,but this does not diminish the value of this work with regard to total neoadjuvant therapy,TNT. Without going into detail about the individual studies,I find this manuscript to be a very good presentation of all previous options for the sophisticated treatment of borderline resectable and locally advanced pancreatic cancer with a comprehensive bibliography and recommend publication in the present version.
Author Response
We appreciate your feedback on our review. Thank you very much for your comments.
Reviewer 4 Report
Comments and Suggestions for Authors
This manuscript about total neoadjuvant therapy (TNT) for localized pancreatic cancer offers a comprehensive analysis of an evolving treatment strategy. The exploration of TNT as an approach to improve surgical success and survival rate in pancreatic ductal adenocarcinoma is crucial, considering the poor prognosis associated with this cancer. The categorization of TNT strategies based on radiotherapy techniques, including conventional chemoradiotherapy, Dutch PREOPANC approach, and various SBRT techniques, provides valuable insights into the current landscape of neoadjuvant treatment options. This structured analysis not only enhances our understanding of these strategies but raises important questions about optimizing treatment sequences to achieve better outcomes.
The discussion on the potential benefits and risks of TNT is particularly commendable. By highlighting the challenges associated with disease progression and surgical complications, the authors present a balanced view of the TNT approach. The comparative analysis of different TNT regimens, including the promising results from the PREOPANC-1 trial and the role of high-dose SBRT, underscores the need for tailored treatment protocols. The manuscript effectively integrates data from multiple studies, providing a clear picture of the efficacy and safety of various TNT strategies. This comprehensive review is a valuable resource for clinicians and researchers seeking to improve treatment outcomes for patients with borderline resectable and locally advanced pancreatic cancer.
In addition, the manuscript's discussion on the future perspectives of TNT, including the potential role of novel imaging techniques and personalized medicine, adds significant importance to the review. The consideration of metabolic imaging and liquid biopsies as tools for evaluating treatment response is forward-thinking and aligns with the precision medicine approach. The emphasis on molecular profiling and targeted therapies further underscores the manuscript's relevance in the context of contemporary oncology research. Overall, this manuscript is a well-written and insightful review contributing significantly to the field of pancreatic cancer treatment. I recommend to accept it.
Author Response

(The authors gave the same response as above.)

Reviewer 5 Report
Comments and Suggestions for Authors
I appreciate the opportunity to review your manuscript titled "Total Neoadjuvant Therapy in Localized Pancreatic Cancer: Is More Better?". The paper addresses significant and clinically relevant questions in the field of pancreatic ductal adenocarcinoma (PDAC). However, I have several major points that need to be addressed to enhance the clarity and impact of your work:
Major points
# The discussion on the general risks and benefits of NAT is unclear. It is not evident whether the authors view patient selection through NAT as a benefit or a demerit. Clarifying this perspective will help readers understand the rationale and implications of using NAT in the context of PDAC.
# For the trials listed in Table 1, please include information on the control groups if they exist. This addition will provide a more comprehensive comparison and context for evaluating the effectiveness of the various TNT strategies discussed.
# Given the already considerable length of the manuscript, the section "Available Tools for the Evaluation of Treatment Response" could be minimized or potentially omitted. This would help to streamline the content and maintain focus on the primary objectives of the paper.
Minor points
# Line 55: “cause sine qua non” this phrase is hard to understand, so that the authors should paraphrase it.
# Line 122: There is duplication of “,”
Comments on the Quality of English LanguageMinor editing/ proof-reading will suffice.
Author Response
We appreciate your feedback on our review.
Major points:
Comment 1: The discussion on the general risks and benefits of NAT is unclear. It is not evident whether the authors view patient selection through NAT as a benefit or a demerit. Clarifying this perspective will help readers understand the rationale and implications of using NAT in the context of PDAC.
Response 1: Thank you very much for your comment. The chapter describes both theroretical advantages and risks of NAT/TNT. The authors tried to "address" concerns raised by potential risks. Overall we believe that the advantages appear to outweight the drawbacks. As such, this was more clearly stated in the section.
Comment 2: For the trials listed in Table 1, please include information on the control groups if they exist. This addition will provide a more comprehensive comparison and context for evaluating the effectiveness of the various TNT strategies discussed.
Response 2: Thank you for your comment regarding table 1. That information is already included in the table. Whenever studies included more than a cohort, the treatment was descreibed and results were included (e.g. Fitekau, 2022, Loka 2021, Pietrsaz 2018...). In addition, whenever the trials were randomised, the term "Rando" was included in the study design column.
Comment 3: Given the already considerable length of the manuscript, the section "Available Tools for the Evaluation of Treatment Response" could be minimized or potentially omitted. This would help to streamline the content and maintain focus on the primary objectives of the paper.
Response 3: Thank you for your comment, that we highly value. Section "Available Tools for the Evaluation of Treatment Response" was shortened to maintain the focus on the primary objectives of the paper.
Minor points:
Comment 1: Line 55: “cause sine qua non” this phrase is hard to understand, so that the authors should paraphrase it.
Response 1: Thank you for your comment. The sentence was paraphrased for a better undestanding.
Comment 2: # Line 122: There is duplication of “,”
Respond e2: Thank you for pointing this out. The extra ",' was removed.